# Cardiorespiratory Fitness, Obesity and Physical Activity in Schoolchildren: The Effect of Mediation

**DOI:** 10.3390/ijerph192316262

**Published:** 2022-12-05

**Authors:** Noelia González-Gálvez, Jose Carlos Ribeiro, Jorge Mota

**Affiliations:** 1Sports Injury Prevention Research Group, Facultad de Deporte, UCAM Universidad Católica de Murcia, 30107 Murcia, Spain; 2Research Center in Physical Activity, Health and Leisure (CIAFEL)—Faculty of Sport—University of Porto (FADEUP) and Laboratory for Integrative and Translational Research in Population Health (ITR), 4200-450 Porto, Portugal

**Keywords:** body mass index, exercise, mediation, schools, fat mass, physical fitness

## Abstract

There are only a few studies that have examined the interaction between physical activity (PA), cardiorespiratory fitness (CRF), and overweightness/obesity in adolescents, and these studies have shown some limitations. The objective of this study was to examine whether the association between PA (moderate–vigorous PA [MVPA], moderate PA [MPA], and vigorous PA [VPA]) and obesity is mediated by CRF. This cross-sectional study was conducted in six middle and high-schools in the Porto district (Portugal), comprising 632 children and adolescents. Fat mass (FM), CRF, MVPA, MPA, and VPA were assessed. The Process macro for SPSS was used. CRF was significantly associated with PA for both children and adolescents. Likewise, FM was negatively and significantly associated with CRF for both children (β = −0.337) and adolescents (β = −0.255). FM was associated with PA in children (MVPA: β = −0.102; MPA: β = −0.118; VPA: β = −0.305) and adolescents (MVPA: β: −0.103; MPA: β = −0.082; VPA: β = −0.204). The mediation analysis showed that the association between PA (MVPA//MPA/VPA) and obesity was mediated by CRF, in both children and adolescents, with a significant value in the Sobel test. Obesity is associated with CRF and MVPA, MPA, and VPA. However, CRF acted as a complete mediator between the association of obesity and PA.

## 1. Introduction

In 2016, more than 18% of children and adolescents aged between 5 and 19 years old were shown to be overweight or obese [1]. This problem is considered a pandemic [2] and is growing worldwide [3]. Overweightness and obesity have been shown to have an association with cardiorespiratory fitness (CRF) [4,5,6,7,8,9] and physical activity (PA) [10,11,12,13,14,15] in children and adolescents. CRF has been identified to be a partial mediator of the association between PA and metabolic syndrome [14], but there are only a few studies that have examined the interaction between PA, CRF, and overweightness or obesity in adolescents [16,17,18,19]. However, these studies have shown some limitations. Some of this research did not directly relate CRF to the rest of the variables since they used a score derived from several fitness tests [6,20]. Some of them used waist circumference (WC) [15] or body mass index (BMI) as a value for obesity [6,20]. However, WC has shown great variability, and repeated training is needed to take this measurement accurately, and BMI does not provide data on body composition [19]. In addition, some of these studies asked for PA [6,17,20], weight, and height [17] with a subjective method (questionnaire), showing a risk of bias [21]. In addition, only one study used a proper statistical analysis to assess the effect of the mediation [16], and the rest of them used multiple linear regression, logistic regression, or analysis of the covariance (ANCOVA) to adjust for mediator variables. However, these multivariate methods did not show the statistical effects due to the mediation.

The association between obesity and CRF, physical fitness, and PA has been reported to be different between children and adolescents [10], and the mediation effect of physical fitness on the association between PA and cardiometabolic risk is different between children and adolescents [14]. Thus, current information on this subject still remains insufficient.

Therefore, the aim of this study was to examine whether the association between PA (MVPA, MPA, and VPA) and obesity is mediated by CRF in both children and adolescents.

## 2. Materials and Methods

### 2.1. Participants

This study was conducted in 6 middle and high-schools in the Porto district and included 632 children and adolescents (girls = 56.01%; boys = 43.99%) aged 13.28 ± 2.47 years old. The sample of children consisted of 345 individuals (aged from 10 to 12 years old; mean: 11.27 ± 0.67), and the sample of adolescents consisted of 287 individuals (aged from 12 to 18 years old; mean: 15.69 ± 1.58).

The RStudio 3.15.0 software was used for the calculations to establish the sample size. The significance level was set at α = 0.05. An estimated error of 1 min/kg was established, and according to the standard deviation set for CRF in previous studies [16,22], a valid sample size for a confidence interval of 95% was 270.41. A total of 632 students completed the trial, which provided a power of 95%, between and within a variance of 0.65 mm min/kg.

### 2.2. Measures

The participants were measured by the same trained researchers in a single session between the hours of 09:00 and 11:00. Anthropometric variables were collected with bare feet and at random. The participants were instructed to wear light clothes. The laboratory temperature was standardized at 24 °C. The Course–Navette test was performed without a previous warm-up. There was a 5 min rest between measurements.

Overweightness and obesity: FM was measured with a bioelectrical impedance portable digital scale (TANITA BF-522 W, Tokyo, Japan). Body mass was measured with a portable digital scale (TANITA BF-522 W, Tokyo, Japan), and height was measured with a SECA 217 stadiometer (SECA, Hamburg, Germany). BMI was calculated with the formula: body mass (kg)/stature (m)^2^ [23].

CRF assessment: The Course–Navette test (20 m shuttle run test) was used to assess CRF, as described elsewhere [24], from a starting speed of 8.5 km h^−1^, as described in the protocol cited. The maximum oxygen consumption (VO_2max_, mL/kg/min) was estimated with the number of laps performed by the participant during the test, using the equation reported by Leger et al. [24].

PA: An ActiGraph accelerometer (GT3X-plus; ActiGraph, Pensacola, FL, USA) and ActiLife software (version 6.11.4; ActiGraph) were used to assess PA. The ActiGraph accelerometer had been previously validated with children [25]. The participants wore an accelerometer for 7 consecutive days, and could only take it off for showering or water-based activities. They wore the accelerometer tightly attached to the hip by an elastic belt on the right side of the body. The recording was considered valid for analysis if the participant wore it for at least 4 days and at least 10 h per day; this criterion was met for all participants. The tri-axial vector magnitude was computed as VM3 = √(VT2 +AP2 + ML2). MPA and VPA intensities were calculated according to the recommended PA vector magnitude cut point. The data derived were interpreted as MPA if the VM3 value ranged from 2690 to 6166; VPA had a VM3 value higher than 6167 [26]. The sampling period was set up for 5 s. This period has been reported to be the most suitable for the spontaneous and intermittent activities of children [27,28].

### 2.3. Procedures

This cross-sectional study was part of a longitudinal study developed in Porto (Portugal), named the AFINA-te Project Study (PA and Nutritional Information for Adolescents). This project was designed as an intervention project to promote nutritional knowledge and PA, with a research grant from the Foundation for Science and Technology [FCOMP-01-0124-FEDER-028619 (PTDC/DTP-DES/1328/2012)].

Ethical approval for this study was obtained from the Ethics Committee of the Faculty of Sports from the University of Porto (Process CEFADE 13/2013), the National Data Protection Commission (process n.6766/2015), and the Regional Section of the Ministry of Education (process 0053200004), and was implemented according to the guidelines for human research of the Helsinki Declaration. All the children, adolescents, and parents/tutors signed an informed consent form. The cross-sectional study design followed the Strobe Statement.

### 2.4. Statistical Analysis

The normality of the data was evaluated using the Kolmogorov–Smirnov test. The differences between groups were analyzed with a t-test, the Mann–Whitney U test, and X2 analyses when necessary. The statistic contingency coefficient post-hoc comparison was applied for 2xn tables, showing the value of the statistic and the *p* value. The maximum expected value was 0.707; a low association was defined if r < 0.3, a moderate association if the r value was between 0.3 and 0.5, and a high association if r > 0.5. To test the associations of the variables, a linear regression was performed, where each variable was individually introduced as an independent variable and as a dependent variable. The analysis of the mediation variables was performed using Process macro for SPSS (SPSS Inc., Chicago, IL, USA). A resampling procedure of 10.000 bootstrap was used for nonparametric variables [29], and the classical Baron and Kenny step regression method [30] was used for parametric variables. The Sobel test was used to test the statistical significance of the mediation effect [31]. Complete mediation was considered when the independent variable was not associated with the dependent variable after the mediator had been controlled for, and partial mediation was considered when the association between the independent and dependent variables was reduced but did not disappear. All the analyses were performed separately for children and adolescents. The statistical analysis was performed using IBM SPSS statistics software (version 24.0). An error of *p* ≤ 0.05 was set.

## 3. Results

### 3.1. Descriptive Variables and Associations between Variables

The descriptive data of the variables and the differences between age groups are shown in Table 1.

CRF was significantly associated with PA for both children (MVPA: β = 0.274; MPA: β = 0.326; VPA: β = 0.775) and adolescents (CRF with MVPA: β = 0.325; MPA: β = 0.259; VPA: β = 0.638).

Likewise, obesity (FM) was negatively and significantly associated with CRF for both children (β = −0.337) and adolescents (β = −0.255). For children, MVPA, MPA, and VPA were associated with FM (MVPA: β = −0.102; MPA: β = −0.118; VPA: β = −0.305), while for adolescents, FM was significantly associated with MVPA (β: −0.103), MPA (β = −0.082) and VPA (β = −0.204).

### 3.2. Mediations Analysis

The effect of CRF mediation is depicted in Figure 1 and Figure 2 for children and adolescents, respectively.

After including the CRF variable in the equations, the association between any PA variable (MVPA/MPA/VPA) and FM in any age group (children/adolescents) was no longer significant. The mediation analysis showed no statistically significant direct or indirect effects once the Sobel test was significant. Thus, CRF acts as a mediator in the association between PA (MVPA/MPA/VPA) and obesity (FM) in both children and adolescents.

## 4. Discussion

The major finding of our study was that obesity was associated with CRF and with PA (MVPA, MPA, and VPA). Likewise, PA intensities (MVPA, MPA, and VPA) were associated with CRF for both children and adolescents.

Our results are in agreement with previous studies that showed an association between obesity and CRF [6,8,15] and PA [10,11,12,16,18,32,33], and between PA and CRF [18,34] in young individuals. For instance, Fogelholm et al. [6], in 15–16 year-old adolescents, showed that CRF was associated with obesity (BMI), while PA was positively associated with fitness. In addition, the authors also argued that PA was more prone to be associated with the fitness index than with obesity indicators, which agrees with our data. Likewise, Tambalis Panagiotakos, Psarra, and Sidossis [12], in a sample of 8–17-year-old children and adolescents, showed that PA was inversely and significantly associated with obesity, and they also found a positive and significant association with CRF. Furthermore, they showed that the association of PA was greater with CRF than with obesity, which is in line with the present study.

In our study, the CRF acted as a complete mediator on the association between PA (MVPA, MPA, and VPA) and obesity. Thus, a high level of CRF would fully protect the participants from obesity regardless of the PA.

To the best of our knowledge, there are only two recent studies that have assessed CRF as a mediator of the association between PA and obesity. However, these studies focused only on girls [16,18]. Both studies showed an association between MVPA and obesity indicators (BMI [16], FM, and abdominal FM [18]) with CRF as a mediator, which agrees with our findings. Considering this, these results should be further discussed. Indeed, other studies also found that PA was associated with cardiometabolic risk, but this association became non-significant when physical fitness was included in the model, and physical fitness could act as a partial mediator in the association between PA and cardiometabolic risk [14]. Thus, our data highlighted the importance of implementing a PA program to improve CRF, with a potential subsequent impact on obesity.

In addition, the results of the present study indicate that VPA was the PA intensity that showed a higher association with CRF. This outcome is in line with previous studies [15,33,35]. For instance, in 2016, Poitras et al. [33] conducted a systematic review to evaluate the relationship between the PA level and health indicators in children and adolescents aged 5–17 years old. The authors pointed out that half of the longitudinal studies that examined different PA levels showed that VPA and MVPA reported a favorable prospective association with obesity, whereas no study reported a prospective association between MPA and adiposity. In addition, other studies showed a significant inverse association between VPA and obesity, while no association was found for MPA [15,35]. Likewise, research focused on cardiovascular disease (CVD) factors found that the level of PA that showed a better association was VPA [14,33,35,36]. These outcomes highlight the importance of intensity when we are looking forward to promoting physical activity throughout life to achieve a healthy lifestyle. Currently, the PA recommendations for children and adolescents refer to a minimum of 60 min per day of MVPA [37]. However, perhaps new PA recommendations for children and adolescents that consider the evidence shown on the importance of VPA are necessary [35], especially when one observes the improvement in CRF and associated metabolic improvements during childhood and adolescence [36].

Nonetheless, our study showed differences according to age groups. Indeed, the mediator effect of CRF seemed to be higher among children than adolescents, which agrees with other studies. For instance, Wisnieski et al. [16] analyzed the mediation effect of CRF between the association of PA and obesity and showed a higher effect for early–mid-puberty than for late puberty. This small decrease in the mediation effect in adolescence as compared to childhood could be explained by the fact that the relationship between weight and fitness decreases after puberty [38]. Likewise, Aires et al. [20], in their longitudinal study, showed that children who performed PA and maintained it over time had better physical fitness in adulthood, regardless of the level of obesity. In agreement with Gammon et al. [38], our study supports the idea that the associations found between CRF, PA, obesity, and puberty are complex and need more research with longitudinal studies.

The present study shows some limitations that must be recognized. First, the cross-sectional design did not result in a cause–effect conclusion. Thus, longitudinal design studies to overcome this limitation are necessary. The method used for assessing CRF was the Course–Navette test. Although this method is valid and reliable for children and adolescents [24] and is used for most studies that are similar, this is an indirect method that has some limitations in its interpretation. In addition, it is necessary to consider critical remarks of other authors related to the CRF estimation that indicate that the 20 m shuttle run test is not a valid test of CRF in boys aged 11–14 years [39].

The main strength of the present study is its novelty. To the best of our knowledge, this is the first study that assessed the effect of the mediation of CRF on the relationship between PA and obesity in girls and boys. There are just two recent studies that performed similar statistical analyses [16,18], and one of them only with girls [16]. The use of an objective measurement of PA, such as an accelerometer [27,28], is a strength of this study, as the majority of similar studies reported PA subjectively [6,17,20].

## 5. Conclusions

We found that obesity is associated with CRF, as well as with MVPA, MPA, and VPA, in both children and adolescents. However, CRF acts as a mediator between the association of obesity and PA (MVPA, MPA, and VPA). Further longitudinal studies should be carried out to analyze this aspect over time.

## Figures and Tables

**Figure 1 ijerph-19-16262-f001:**
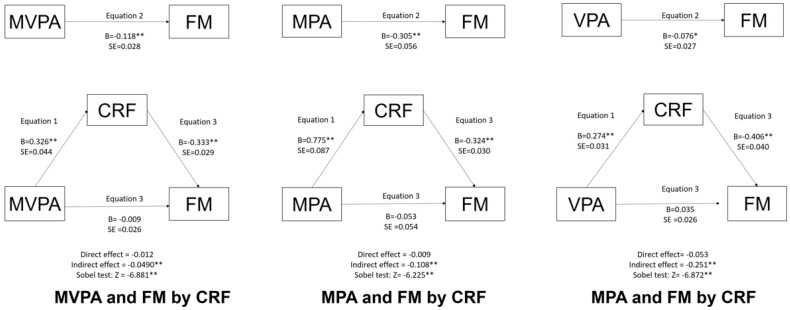
Mediation of MVPA/MPA/VPA and FM by CRF for children. Legend: ** *p* ˂ 0.001; * *p* ˂ 0.05; MVPA = moderate to vigorous physical activity; MPA = moderate physical activity; VPA = vigorous physical activity; CRF = cardiorespiratory fitness; FM = fat mass.

**Figure 2 ijerph-19-16262-f002:**
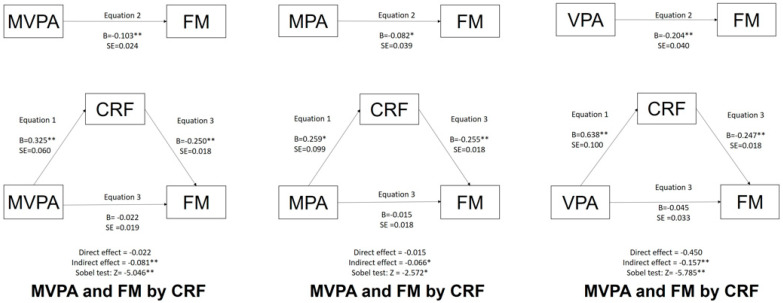
Mediation of MVPA/MPA/VPA and FM by CRF for adolescents. Legend: ** *p* ˂ 0.001; * *p* ˂ 0.05; MVPA = moderate to vigorous physical activity; MPA = moderate physical activity; VPA = vigorous physical activity; CRF = cardiorespiratory fitness; FM = fat mass.

**Table 1 ijerph-19-16262-t001:** Characteristics of the study sample and differences between age groups (children vs. adolescents).

Outcome	Total (*n* = 632)	Children (*n* = 345)	Adolescents (*n* = 287)	*p*
Age (years)	13.28 ± 2.49	11.27 ± 0.67	15.69 ± 1.58	˂0.001
Weight (kg)	53.33 ± 14.17	47.31 ± 12.03	60.57 ± 13.13	˂0.001
Height (cm)	156.57 ± 11.11	149.85 ± 7.58	164.64 ± 9.14	˂0.001
BMI (kg/m^2^)	21.49 ± 4.02	20.88 ± 4.12	22.22 ± 3.77	˂0.001
Normal weight (%(*n*))	81.17 (513)	81.16 (280)	81.18 (233)	0.485
Overweight (%(*n*))	16.14 (102)	16.81 (58)	15.33 (44)	
Obese (%(*n*))	2.69 (17)	2.03 (7)	3.48 (10)	
FM (%)	22.55 ± 8.69	23.92 ± 8.62	20.91 ± 8.50	˂0.001
Course–Navette (number of laps)	4.91 ± 1.8	4.2 ± 1.26	5.38 ± 1.98	˂0.001
CRF (mL/kg/min^−1^)	33.00 ± 19.50	26.64 ± 14.26	40.64 ± 22.06	˂0.001
MVPA (min/day)	43.50 ± 21.59	41.26 ± 22.15	46.19 ± 20.60	0.004
MPA (min/day)	32.04 ± 14.77	32.71 ± 16.14	31.23 ± 12.92	0.209
VPA (min/day)	11.46 ± 10.60	8.55 ± 8.01	14.97 ± 12.19	˂0.001

Legend: BMI: body mass index; kg: kilograms; cm: centimeters; FM: fat mass; CRF: cardiorespiratory fitness; MVPA: moderate–vigorous physical activity; MPA: moderate physical activity; VPA: vigorous physical activity; min: minutes.

## Data Availability

Not applicable.

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
