# Peer review of "Cardiorespiratory Fitness, Obesity and Physical Activity in Schoolchildren: The Effect of Mediation"

_ijerph, 2022, doi:10.3390/ijerph192316262_

Round 1

Reviewer 1 Report

The topic taken up by the Authors is interesting and up-to-date.

I have some comments and suggestions:

If maximum oxygen consumption (VO2max, mL / kg / min) was estimated with the number of laps performed by the participant during the test, using the equation reported by Leger et al, then in table 1 please provide the number of laps covered

• For the sake of clarity of work, it should be specified whether the 20m SRT test began with the running speed 8 km/h as stated on the website https://www.hsnstore.eu/blog/sports/running/course-navette/ or from the speed of 8.5 km/h - this should be explained in more detail in the V02max measurement and estimation procedure.

• The participants wore an accelerometer for 7 consecutive days, and could only take it off for showering or water-based activities. They wore the accelerometer tightly attached to the hip by an elastic belt on the right side of the body. The recording was considered valid for analysis if the participant wore it for at least 4 days and at least 10 hours per day.

If these were the assumptions of inclusion and exclusion from the study, it is necessary to state how many ultimately did not meet this criterion or that they met all of them.

I believe that the discussion should refer to the critical remarks of other authors related to the CRF estimation, eg Welsman, J .; Armstrong, N. The 20 m Shuttle Run Is Not a Valid Test of Cardiorespiratory Fitness in Boys Aged 11–14 Years. BMJ Open Sport - Exerc. Med. 2019, 5 (1), e000627. https://doi.org/10.1136/bmjsem-2019-000627 Especially that the average estimated VO2max results presented in this study seem low.

Author Response

The topic taken up by the Authors is interesting and up-to-date.

  • Thank you very much for your kind words.

I have some comments and suggestions:

  • If maximum oxygen consumption (VO2max, mL / kg / min) was estimated with the number of laps performed by the participant during the test, using the equation reported by Leger et al, then in table 1 please provide the number of laps covered
  • Thank you for the considerations. It is included.

  • For the sake of clarity of work, it should be specified whether the 20m SRT test began with the running speed 8 km/h as stated on the website https://www.hsnstore.eu/blog/sports/running/course-navette/ or from the speed of 8.5 km/h - this should be explained in more detail in the V02max measurement and estimation procedure.
  • Thank you for the consideration. The follow sentence is included: “from starting speed of 8.5 km h-1 as described in the protocol cited”.

  • The participants wore an accelerometer for 7 consecutive days, and could only take it off for showering or water-based activities. They wore the accelerometer tightly attached to the hip by an elastic belt on the right side of the body. The recording was considered valid for analysis if the participant wore it for at least 4 days and at least 10 hours per day.

If these were the assumptions of inclusion and exclusion from the study, it is necessary to state how many ultimately did not meet this criterion or that they met all of them.

  • Thank you for the comment. The follow sentence has been included: “this criterion was meet for all participant”.
  • I believe that the discussion should refer to the critical remarks of other authors related to the CRF estimation, eg Welsman, J .; Armstrong, N. The 20 m Shuttle Run Is Not a Valid Test of Cardiorespiratory Fitness in Boys Aged 11–14 Years. BMJ Open Sport - Exerc. Med. 2019, 5 (1), e000627. https://doi.org/10.1136/bmjsem-2019-000627Especially that the average estimated VO2max results presented in this study seem low.
  • Thank you for the considerations. The follow sentence has been included “In addition, it is necessary to consider critical remarks of other authors related to the CRF estimation that indicate that 20 m Shuttle run test is not a valid test of CRF in boys aged 11-14 years.”

Reviewer 2 Report

The study represented that CRF could be a mediator in obese adolescents and children as well as PA. The novelty of the research could fulfill the gaps from previous studies in which limitations such as only girls were recruited and statistical issues were presented. 

The introduction part was clear and direct to the points although relatively short paragraph.

The protocol and parameter measurements were clearly described.

The results showed a link between CRF and each PA via equations that were appropriate.

The statistical analysis was decent.

The discussion and conclusion were clearly mentioned in the study.

Author Response

  • Thank you very much for your review and time invested in this review.